# Enhanced the Thermal Conductivity of Polydimethylsiloxane via a Three-Dimensional Hybrid Boron Nitride@Silver Nanowires Thermal Network Filler

**DOI:** 10.3390/polym13020248

**Published:** 2021-01-13

**Authors:** Zhengqiang Huang, Wei Wu, Dietmar Drummer, Chao Liu, Yi Wang, Zhengyi Wang

**Affiliations:** 1Sino-German Joint Research Center of Advanced Materials, East China University of Science and Technology, Shanghai 200237, China; Y30180265@ecust.mail.edu.cn (Z.H.); Y12190025@ecust.mail.edu.cn (C.L.); wang_yi@ecust.edu.cn (Y.W.); Y20170049@ecust.mail.edu.cn (Z.W.); 2Institute of Polymer Technology, Friedrich-Alexander-University Erlangen-Nuremberg (FAU), 91058 Erlangen, Germany; drummer@lkt.uni-erlangen.de

**Keywords:** AgNWs, BN, hybrid filler, thermal conductive, polydimethylsiloxane

## Abstract

In this work, polydimethylsiloxane (PDMS)-based composites with high thermal conductivity were fabricated via a three-dimensional hybrid boron nitride@silver nanowires (BN@AgNWs) filler thermal network, and their thermal conductivity was investigated. A new thermal conductive BN@AgNWs hybrid filler was prepared by an in situ growth method. Silver ions with the different concentrations were reduced, and AgNWs crystallized and grew on the surface of BN sheets. PDMS-based composites were fabricated by the BN@AgNWs hybrid filler added. SEM, XPS, and XRD were used to characterize the structure and morphology of BN@AgNWs hybrid fillers. The thermal conductivity performances of PDMS-based composites with different silver concentrates were investigated. The results showed that the thermal conductivity of PDMS-based composite filled with 20 vol% BN@15AgNWs hybrid filler is 0.914 W/(m·K), which is 5.05 times that of pure PDMS and 23% higher than the thermal conductivity of 20 vol% PDMS-based composite with BN filled. The enhanced thermal conductivity mechanism was provided based on the hybrid filler structure. This work offers a new way to design and fabricate the high thermal conductive hybrid filler for thermal management materials.

## 1. Introduction

With the gradual development of micro-miniaturization, high integration density, high power, and multi-function in electronic and electrical industry, the heat accumulation in the field of electronic packaging has become a serious problem [1,2]. The serious heat accumulation, which cannot be dissipated in the use of electronic products, will greatly affect the service effect and life of electronic components, which has become an obstacle for the development of the next generation of electronic components [3,4]. Therefore, it is necessary to design and prepare high conductivity thermal interface materials to eliminate the accumulated heat [5].

Flexible thermal conducting polymers, due to their unique flexibility and stable thermal conduction network, can still maintain good thermal conduction performance when subjected to tensile and bending stress. They were considered to overcome many weaknesses of traditional thermal conducting polymers and are widely used as thermal interface materials [6,7,8,9,10]. Polydimethylsiloxane (PDMS) is often used as the matrix material of thermal interface materials because of its excellent properties, such as flexibility, low cost, chemical inertness, water resistance, and flame retardant [10,11,12,13,14]. However, the intrinsic thermal conductivity of PDMS (about 0.2 W·m^−1^·K^−1^) is not ideal to meet the needs of the electronic industry [15,16,17,18,19]. Therefore, it is necessary to improve the thermal conductivity of PMDS to expand its potential applications.

In particular, because of its excellent thermal conductivity and electrical insulation, hexagonal boron nitride (h-BN) is considered as a potential thermal conductive filler, also known as “white graphite” [20]. Lin et al. stripped BN through liquid phase and blended it with epoxy resin. The addition of 5 wt% enhanced the thermal conductivity of epoxy by 115% [21]. As a two-dimensional material with a few-layers structure, fewer contact points between each other at low filler concentrates leads to a low thermal conductivity. It is difficult to form a heat conduction path in PMDS-based composites. Therefore, a high load (>30 vol%) is usually required to form a three-dimensional thermal network in the matrix [16,22]. However, the mechanical properties of the material will be seriously reduced. And more, the forming process will become difficult.

To the best of our knowledge, one-dimensional thermal conductive materials and two-dimensional thermal conductive materials can often show effective synergistic effects in improving polymer thermal conductivity [23,24]. Chang et al. assembled high-quality ball milling exfoliated graphite nanosheets (BMEGN) and silver nanowires (AgNWs) on a flexible polydimethylsiloxane (PDMS) substrate. When the loading of AgNWs thermal film is 2.0 mg/mL, the in-plane thermal conductivity of the composite material is significantly increased to 29.2 W/(m·K) [25]. Unique connected structure fillers were also reported, which were fabricated by many methods, such as grafting and so on [18,26]. Compared with the traditional single fillers or mixed fillers, hybrid thermal conductive composites can often show better thermal conductivity, resulted from that the synergistic effect of hybrid fillers. The conduction network can be optimized by three-dimensional constructed structure. A higher thermal conductivity of composites can be obtained at a lower filling amount [27,28,29]. It was important to obtain high thermal conductivity under the condition of good mechanical properties and molding processing properties.

Inspired by three-dimensional constructed structure, a new type of hybrid boron nitride@silver nanowires (BN@AgNWs) thermal network filler was designed and fabricated by in situ growth method here. AgNWs with different concentrates grew on the surface of boron nitride to form BN@AgNWs hybrid filler. The BN@AgNWs/PDMS composite was produced by hot pressing method. The effect of AgNWs with different contents on thermal conductivity PDMS-based composites was investigated. The results showed that the thermal conductivity of BN@15AgNWs/PDMS is higher than that of BN/PDMS significantly. This is mainly due to the good compatibility between the silver nanowires and PDMS matrix, and the “silver bridge” formed by connecting the silver nanowires between the boron nitride sheets optimizes the three-dimensional heat conduction network. This work can offer a new idea to design and fabricate the high thermal conductive composites.

## 2. Experimental Section

### 2.1. Materials

The BN powder (50 μm, purity > 99%) was purchased from Jingyi Ceramic Technology Co., Ltd., Zibo, China. The chloroplatinic acid (purity > 99%) was purchased from Maclean Biochemical Technology Co., Ltd., Shanghai, China. Silver nitrate (AR, ≥99.8%) and glycol (AR, >99.0%) were provided by Titan Technology Co., Ltd., Shanghai, China. Polyvinylpyrrolidone (PVP, average molecular weight 58,000), Karstedt catalyst (1000 parts per million), and sodium hydroxide (purity > 96%) were purchased from Sigma-Aldrich Technology Co., Ltd., Shanghai, China. Polymethylhydrosiloxane (0.8 wt% a-H), Vinylterminated poly (dimethylsiloxane) (average molecular weight 17,300 and 116,500), and VMQ silicone resin (average molecular weight 62,700) were provided by Jiande Polymerization New Material Co., Ltd., Jiande, China. 1-Ethynylcyclohexanol was purchased from Maclean Biochemical Technology Co., Ltd., Shanghai, China.

### 2.2. Preparation of BN@AgNWs

The preparation method of BN@AgNWs is shown in Figure 1. The hexagonal boron nitride (h-BN) was treated with 5 M NaOH solution [30]. Firstly, 50 g h-BN was dispersed in 1000 mL NaOH solution. After reaction at 120 °C for 48 h, it was taken out and filtered, washed with deionized water to neutral, and then dried in an oven. The sample was recorded as h-BN. Then, the reactive Pt-sites were introduced onto the surface of f-BN. In brief, 80 mL glycol and 2 g f-BN were added into the three-port flask. After condensation and reflux at 160 °C for 1 h, a 20 mL (0.096 mM) (glycol as solvent) chloroplatinic acid solution was dripped and stirred continuously. The relevant reactions are revealed in Equations (1) and (2).

2HOCH_2_—CH_2_OH → 2H_2_O + 2CH_3_CHO
(1)

H_2_PtCl_6_ + 4CH_3_CHO → 2CH_3_CO—COCH_3_ + Pt + 6HCl
(2)


The AgNWs were obtained by reducing silver nitrate with the addition of PVP, which act as morphology controlling agents. Because silver and Pt have similar lattice energy, the reduced silver can grow into silver nanowires on the surface of Pt nanoparticles at the presence of PVP [31]. PVP (0.72 M glycol as solvent) and silver nitrate (0.2 M glycol as solvent) growth solution were synchronously added into the three-port flask, and reacted at 160 °C for 1 h, then naturally cooled to room temperature. The samples were washed five times with ethanol and deionized water, and then dried in the oven at 100 °C for 12 h. The obtained powder is marked as BN@xAgNWs, where x represents the mass loading of AgNWs. The corresponding recipe table was listed in Table 1.

### 2.3. Preparation of the BN@xAgNWs/PDMS Composites

The preparation of PDMS composites refers to the previous work of our research group [18]. The calculated and weighed BN@xAgNWs hybrid filler is added to the PDMS matrix. The mixture is evenly stirred for 20 min at 1000 rpm by mechanical stirring. The samples were formed by hot pressing technique. First, the uniformly mixed samples were poured into the mold, solidified under 10 Mpa at 150 °C for 10 min, and then cooled to room temperature to obtain composite samples. The volume fraction of the hybrid filler changes from 0 vol% to 20 vol% with a 5 vol% interval.

### 2.4. Characterization

The field emission scanning electron microscopy (SEM, geminisem 500) was used to observe the microstructure of BN@AgNWs. Before SEM testing, the sample was sprayed with a layer of gold film in the vacuum chamber. High magnification micrographs were obtained with the accelerating voltage of 15.0 kV. The crystal structure of the hybrid filler was characterized by X-ray diffraction measurements, scanning at 0.02°/s in the 2θ range of 5–90° using a X-ray Diffractometer (D8 Advance, Bruker Co., Billerica, MA, USA) with Cu- Kα radiation (λ = 0.154 nm,) at 40 kV and 40 mA. The hot wire method was used to test the thermal conductivity (λ) of the composite material. The test sample is a thin plate of 60 mm × 40 mm × 2 mm after cleaned. The same samples with five times were tested at 25 °C using TC3000E (China Xi’an Xiaxi Electronic Technology Co., Ltd., Xi’an, China). After each test, the sample needs to be left to cool down before the next test to reduce errors. The hot wire method is mainly used to test the thermal conductivity (λ) of solid polymers with temperature. The principle of hot-wire measurement of thermal conductivity can be explained in this way: Place a thin thermal resistance wire between two solid thin samples. Then, fix it with silicone grease sheets on both sides. At the same time, a 500 g weight is placed on the silicone grease sheet to provide a constant pressure. When the current is stable, the resistance wire generates heat at a constant rate and then transfers to the contact sample. The λ value of the material can be determined by the heat value (*Q*) per unit length of the resistance wire and the value of the resistance wire. The natural logarithm relationship between the temperature change slope (ΔT) and time is:(3)$$\lambda =\frac{Q}{4\pi }/\frac{\mathrm{dT}}{\mathrm{dlnt}}$$

## 3. Results and Discussion

The SEM images of AgNWs and BN hybrid fillers with different mass ratios are shown in Figure 2, AgNWs can be seen on the surface and the periphery of BN, and they have a better dispersion state at low feed ratios. With the increase in the quality of AgNO_3_, the number of AgNWs continues to increase. In Figure 2e, AgNWs have begun to agglomerate. In Figure 2f, AgNWs have already agglomerated and the dispersion of them is poor. The agglomeration will have a bad influence on the thermal conductivity. Because, for filler-filled composites, its thermal conductivity is substantially affected by the density of filler pathways. The higher the dispersion of filler, the higher density of filler channels. This means that more robust filler networks are constructed, which can help the fabrication of high thermal conductivity composites. When the content of AgNWs is low, as shown in Figure 2b,c, only a small amount of AgNWs can overlap between the BN sheet and the sheet. The small amount of AgNWs contributes little to the formation of the heat conduction path, mainly the heat conduction path formed by BN. With the increase of AgNWs content as shown in Figure 2d,e, it can be clearly seen that AgNWs grown on the surface of BN are uniformly distributed among BN.

The BN@xAgNWs hybrid fillers are connected to each other to form a silver bridge, resulting in more heat conduction paths formed between the fillers. When AgNWs contents continue to increase, as shown in Figure 2f, AgNWs no longer appear in the form of a silver bridge. Instead, it is free to accumulate on the surface of BN to form a network of AgNWs, which is obviously not in line with the purpose of the research.

The XRD pattern of untreated BN and BN@AgNWs was shown in Figure 3. The main peaks at 26.5°, 42.8°, 43.6°, 50.4°, 55.0°, 76.1°, and 82.3° are from the characteristic peaks of h-BN. Compared with pure BN, BN@xAgNWs has 5 new diffraction peaks. The diffraction peak are located at 38.19°, 44.50°, 65.59°, 77.47°, and 81.6°, corresponding to the diffraction crystal planes (111), (200), (220), (311), and (222) of AgNWs. It revealed that the silver nanowires on the BN surface have face-centered cubic structures. In addition, the XRD pattern shows that the intensity of the diffraction peak of the (111) crystal plane is much higher than that of other crystal planes, which proves that the growth speed of the silver crystal on a certain crystal plane is faster. The growth speed of the crystal along the (111) crystal plane is much greater than other crystal planes. This indicated that the (111) crystal plane of AgNWs was the predominant growth orientation. This is mutually corroborated with the SEM image. Therefore, it can be preliminarily determined that the BN@xAgNWs hybrid filler has been successfully synthesized.

The element status of the BN@xAgNWs hybrid filler was analyzed by XPS. As the N1s peak before and after the AgNWs modification show in Figure 4, the core energy peak shifted from 397.6 eV to 397.5 eV. This shift of the core energy peak indicates the chelating interaction between BN and the Pt nanoparticle sites of AgNWs. The XPS and XRD patterns, which can prove that BN@AgNWs was successfully synthesized on the surface of BN and combined with chemical interaction.

The thermal conductivity of materials reflects the heat transfer efficiency through the heat transfer rate. As shown in Table 1, the synthesized five BN@xAgNWs and untreated BN were respectively filled into the PDMS matrix with a filler ratio of 5% vol to obtain five BN@xAgNWs/PDMS composite materials and BN/PDMS composite materials. The curves in Figure 5a showed thermal conductivity of PDMS with 5 vol% BN@xAgNWs hybrid fillers by measuring the composite materials. When x = 15, the thermal conductivity of BN@15AgNWs/PDMS reaches its maximum value. The overall thermal conductivity shows a trend of first rising and then falling. When x = 25, the thermal conductivity is even lower than that of the pure BN/PDMS composite.

The SEM results also indirectly explained the changes of thermal conductivity above. When the AgNWs content is low, only a small part of BN and AgNWs are connected. An effective heat conductive network cannot be formed. The effect on the formation of the new heat conduction path is limited, and the thermal conductivity is not significantly improved. When x = 15, as shown in Figure 2d, it can be clearly seen that BN@AgNWs are basically evenly distributed and overlap each other, forming a silver bridge between the BN sheets. The thermal conductivity reaches the maximum value. Therefore, it can be concluded that when x = 15, the best thermal conductivity network is formed inside the thermally conductive materials.

Additionally, the subsequent selection of BN@xAgNWs as the filler continues to increase the addition amount as shown in Figure 5b, starting from 5 vol% and preparing BN@15AgNWs/PDMS and BN/PDMS thermally conductive composite materials at 5% intervals.

Figure 6 shows the distribution of the filler in the PDMS matrix and the formation of the thermal network. As shown in Figure 6a, when the BN loading content is low, BN is evenly distributed in the PDMS. The filler forms a sea-island structure in the matrix, and the fillers cannot make contact to form an effective heat conduction network, leading to a low heat conduction performance. As the BN content increased in Figure 6b, the BN sheets could be contacted with each other, effectively forming a thermal conductive network inside the substrate. The thermal conductivity of the materials has been significantly improved. However, due to the poor compatibility of BN and PDMS matrix, more cavities are formed between the contact surfaces, which obviously increases the scattering of phonons and affects the improvement of the thermal conductivity of the composite materials. However, as shown in Figure 6c,d, the presence of AgNWs forms a silver bridge between BN and BN particles [32,33]. Due to the existence of these silver bridges, a new path of the BN-AgNWs-BN structure is constructed between boron nitride and boron nitride. The BN-AgNW-BN phonon transmission path was introduced in the composite systems, which increases the possibility of heat conduction.

Compared with BN, BN@15AgNWs has better binding effect with PDMS matrix due to the existence of AgNWs. In the process of hot pressing, BN@15AgNWs forms an orientation inside the composite materials. With the existence of AgNWs in filler materials, a new thermal conduction network based on BN@15AgNWs was formed, which effectively reduced the thermal conductive gap of PDMS and reduced phonon scattering to improve the thermal conductivity. As shown in Figure 5b, the thermal conductivity of the composite materials gradually increases with filler contents’ increased. However, the thermal conductivity of BN@15AgNWs/PDMS increases faster, compared from BN/PDMS. When the filler content reaches 20%vol, the thermal conductivity of the composite material is up to 0.91 W·m^−1^·k^−1^, which is 23% higher than that of the BN/PDMS composite materials with the same addition contents, and is compared with pure PDMS (0.18 W·m^−1^·k^−1^) increased by 505%.

Figure 7 showed tensile strength and elongation at break of thermally conductive BN/PDMS and BN@15AgNWs/PDMS composites. Tensile tests were performed on BN/PDMS and BN@15AgNWs/PDMS, respectively, in Figure 7a. With filler contents increased, the tensile strength of BN/PDMS and BN@15AgNWs/PDMS decreased. However, the decline rate of BN@15AgNWs/PDMS was significantly lower than that of BN/PDMS. When the filler content was 20 vol%, it was higher than that of BN/PDMS. The tensile strength of PDMS increased by 25.4%. The filling amount was increased, namely the AgNWs content increases, BN@15AgNWs/PDMS composites had a higher tensile strength. Additionally, the elongation at break has a similar trend. As the filler contents increased, the elongation at break also gradually decreased. The compatibility of AgNWs and PDMS is better than that of BN and PDMS. With the increase of the filler loading, the mechanical properties of BN@15AgNWs/PDMS are better than BN/PDMS, which may be due to the AgNWs grown on the surface of BN@AgNWs improving the interface interaction between the filler and the matrix. Compared with BN/PMDS, the new hybrid filler BN@15AgNWs synthesized in this study is used to fill PDMS to improve thermally conductive and mechanical properties of composite systems. This work can offer a new understanding that these BN@15AgNWs/PDMS with excellent thermal conductivity and mechanical properties have important application potential in the field of thermal management materials.

## 4. Conclusions

In summary, PDMS-based composites with BN@AgNWs hybrid filler were fabricated successfully, and their thermal conductivity performance was investigated. The thermal conductive BN@AgNWs hybrid filler was prepared by an in situ growth method. The results showed that the thermal conductivity of PDMS-based composite filled with 20 vol% BN@15AgNWs hybrid filler is 0.914 W/(m·K), which is 5.05 times that of pure PDMS and 23% higher than the thermal conductivity of composite filled with the same loading of original BN fillers. The tensile strength and elongation at break of BN@15AgNWs/PDMS composites have been improved, compared with thermally conductive BN/PDMS composites. Our results show that the combination of 1D AgNWs and 2D BN fillers can construct better filler networks than single BN fillers, which gives a new insight for endowing composites with optimized 3D filler construction via incorporating hybrid fillers. Overall, this work provides a promising method for the synthesis of a highly thermally conductive hybrid filler suitable for the manufacturing of high-performance thermal interface material for electronics.

## Figures and Tables

**Figure 1 polymers-13-00248-f001:**
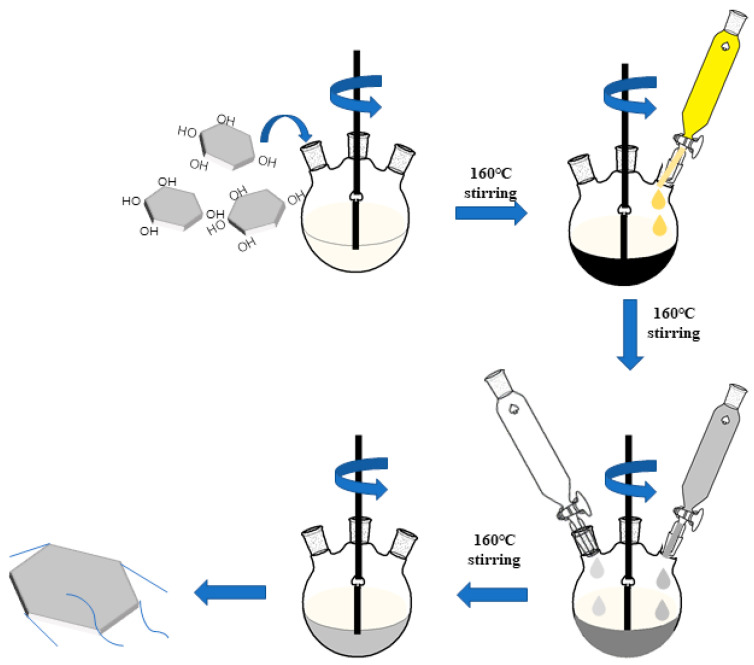
Schematic illustration of the preparation procedures for BN@AgNWs (boron nitride@silver nanowires). (Yellow: Chloroplatinic acid, gray: Silver nitrate solution, transparent: Polyvinylpyrrolidone (PVP) solution).

**Figure 2 polymers-13-00248-f002:**
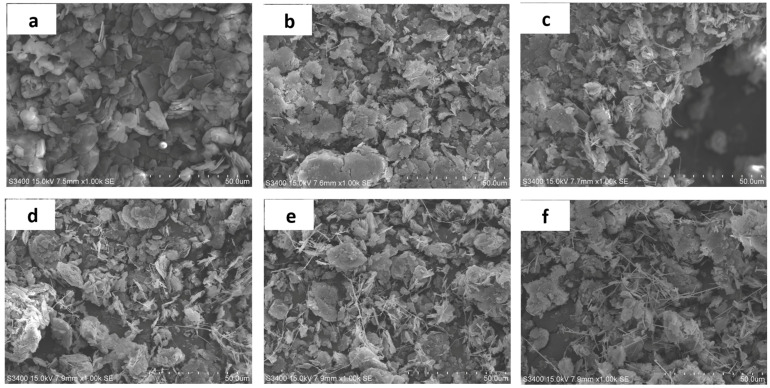
SEM images of (**a**) pure BN; (**b**) BN@5AgNWs; (**c**) BN@10AgNWs; (**d**) BN@15AgNWs; (**e**) BN@20AgNWs; (**f**) BN@25AgNWs.

**Figure 3 polymers-13-00248-f003:**
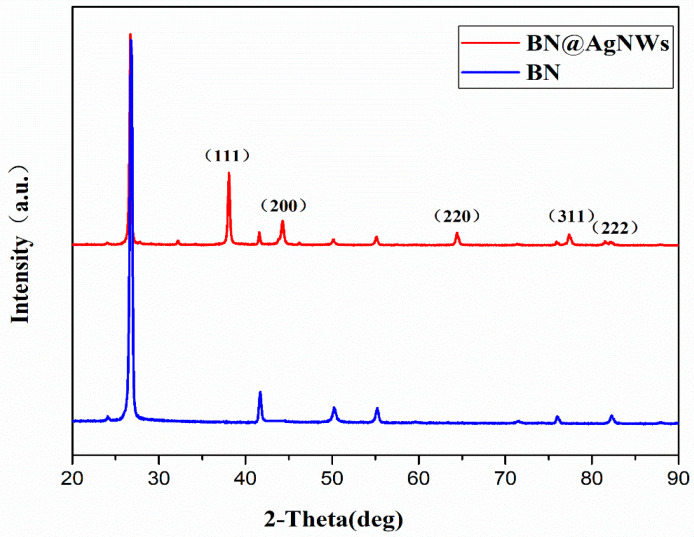
XRD patterns of BN and BN@AgNWs hybrid fillers.

**Figure 4 polymers-13-00248-f004:**
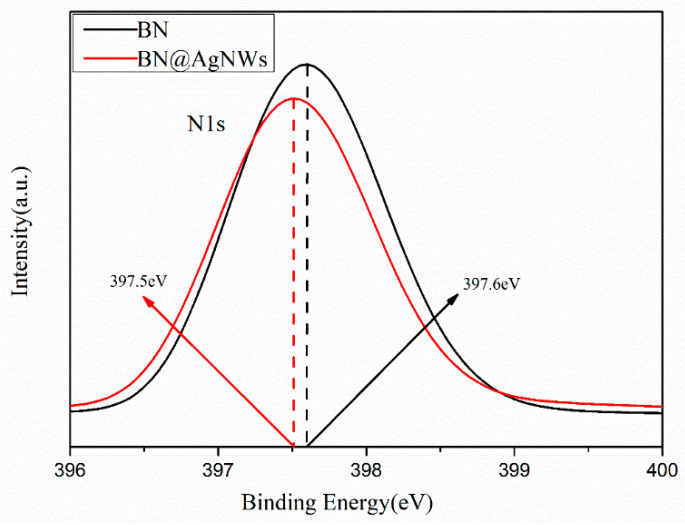
XPS N1s spectra of the BN and BN@AgNWs.

**Figure 5 polymers-13-00248-f005:**
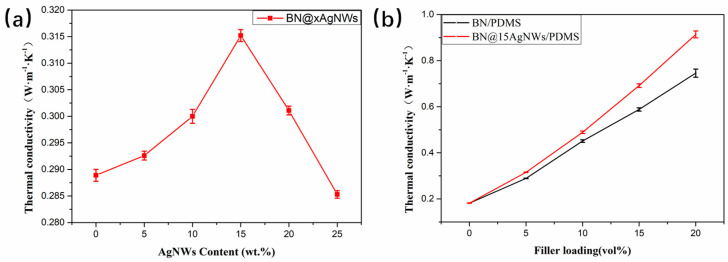
(**a**) Thermal conductivity of polydimethylsiloxane (PDMS) with 5 vol% BN@xAgNWs hybrid fillers and (**b**) thermal conductivity of BN@15AgNWs PDMS with different filler content.

**Figure 6 polymers-13-00248-f006:**
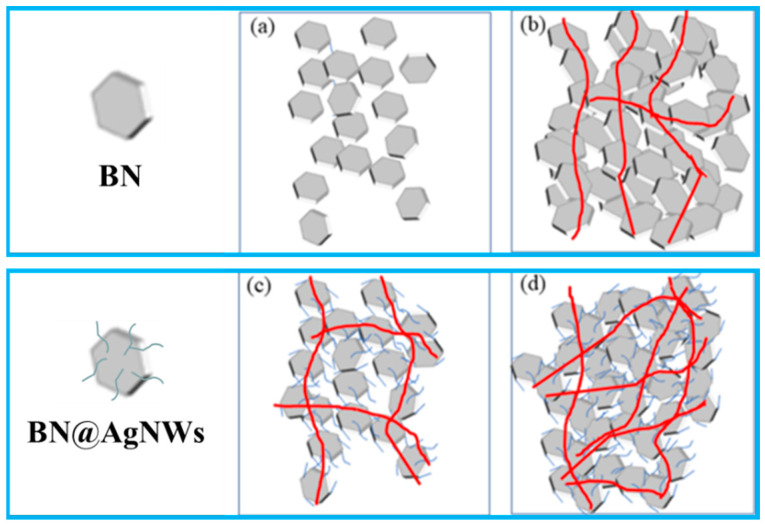
Schematic diagram of the dispersion of BN and BN@15AgNWs in PDMS matrix: (**a**) Low filler content BN/PDMS, (**b**) high filler content BN/PDMS, (**c**) low filler content BN@15AgNWs/PDMS, and (**d**) high filler content BN@15AgNWs/PDMS.

**Figure 7 polymers-13-00248-f007:**
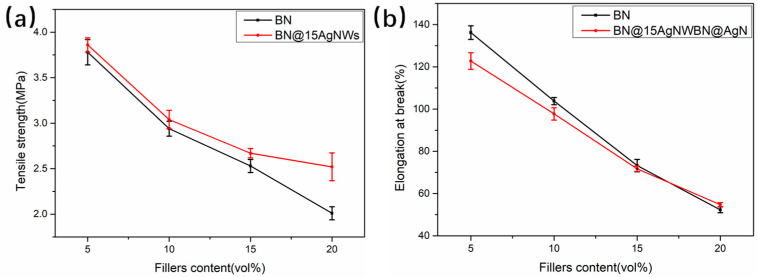
Tensile strengths (**a**) and elongation at break (**b**) of BN/PDMS and BN@15AgNWs/PDMS as a function of variant fillers contents.

**Table 1 polymers-13-00248-t001:** Formula for BN@xAgNWs.

Sample	BN (g)	H_2_PtCl_6_ (mL)	EG (mL)	AgNO_3_ (g)	PVP (g)
BN@5AgNWs	2.00	20.00	80.00	0.157	0.37
BN@10AgNWs	2.00	20.00	80.00	0.315	0.74
BN@15AgNWs	2.00	20.00	80.00	0.472	1.11
BN@20AgNWs	2.00	20.00	80.00	0.628	1.48
BN@25AgNWs	2.00	20.00	80.00	0.785	1.85

## Data Availability

The [DATA TYPE] data used to support the findings of this study are available from the corresponding author upon request.

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
