# Peer review of "Enhanced the Thermal Conductivity of Polydimethylsiloxane via a Three-Dimensional Hybrid Boron Nitride@Silver Nanowires Thermal Network Filler"

_polymers, 2021, doi:10.3390/polym13020248_

Round 1

Reviewer 1 Report

Comments to the authors

This work investigated enhancement on the thermal conductivity of polydimethylsiloxane filled with three-dimensional hybrid boron nitride/silver nanowires thermal network filler. The authors used optimal amount of the fillers to improve the thermal conductivity of the composite.  The topic is interesting and will provide an insight for future research works. Taking this into consideration, I recommend the authors to consider the following points.

  1. In the introduction, the background information on each component of the composite was provided but there is no enough reference work related to the effect of BN and Ag NWs on the thermal conductivity of previously fabricated composites. Please include and cite previous works related to this research work.
  2. In the experimental, please provide the hot-press processing temperature. If any.
  3. Figure 2. Is complicated to understand for the readers and doesn’t necessary align with the discussion. I recommend including the SEM image of pure BN which makes it easy to compare the difference and improve the readability.
  4. On page 5 line 144, the authors stated that “The agglomeration will have a bad influence on the thermal conductivity.” What is the effect of filler agglomeration on the thermal conductivity of the composite? Please discuss.
  5. On page 8 line 248, the authors mentioned “The main reason for the improvement of mechanical properties under higher filler content…”. Are you sure? The characterization done is not enough to conclude like this. Try to include the Young’s modulus value from the stress Vs strain curve or perform additional mechanical property analysis.
  6. Finally, I recommend the authors to go through the manuscript again to check type errors, word choice and some grammatic errors.

Based on the comments, this manuscript can be recommended for publication in ‘Polymers’ after minor revision.

Reviewer 2 Report

In this contribution by Huang et al., the authors synthesized BN@AgNW materials and incorporated them into a polydimethylsiloxane matrix. After that, the mechanical and thermal properties of the composite were established. The findings are relevant to the journal of Polymers as the research completely fits its aims and scope. However, certain issues must be addressed before the paper can be recommended for publication. Please find the suggestions below:
1) Headlines should not be separated from the corresponding sections (Line 88)
2) What is "purity>99"?
3) "The chloroplatinic acid (purity>99) was friendly purchased from Maclean Biochemical Technology Co., Ltd, Shanghai (China)." - what does it mean?
4) "Polyvinylpyrrolidone (PVP, molecular weight 58000)" - what is the unit?
5) "Karstedt catalyst (1000 ppm)" - ppm per what? What is the medium?
6) "sodium hydroxide (>96%)" - what is the unit?
7) "Vinylterminated poly (dimethylsiloxane) and VMQ silicone resin" - what is the MW and other parameters?
8) " The hexagonal boron ni- 91
tride (h-BN) was treated with 5 M NaOH solution firstly" - no mention of the ratio.
9) "After 48 h reaction at 120 °C" - how was an aqueous solution kept at 120C without applying any form of pressure
10) "Then, the seeds were seeded with chloroplatinic acid on the surface of f-BN" - no mention of the ratio either, treatment conditions, etc.
11) Unknown acceleration voltage for SEM.
12) Overall, the experimental section is not written with sufficient scientific rigor, which does not enable the reproduction of the results. As a consequence, the possible impact of this work is low. Please carefully go through the article and include all the necessary details to make these findings reproducible by others. Only examples of mistakes were shown above.
13) Fig. 1 - increase its size because it is not clear what is on the image. Moreover, indicate on the scheme what are these yellow, transparent, and grey solutions.
14) "The same samples with five times were tested at 25°C" - did the authors test if different samples give similar results?
15) Captions should not be separated from plots (Line 137)
16) Not all XRD features are described in the plot of patterns in Fig. 3.
17) A serious concern is the lack of error bars in Fig. 5. and 7, which puts doubt if these results should be interpreted. Please correct this shortcoming.
18) The description of the impact of this work and future outlook should be given in the Conclusions section.

Round 2

Reviewer 2 Report

Thank you for following the suggestions. However, the formatting of the article after the revision is very strange as there is a lot of empty space. Please correct it at the proofing stage. The article can be accepted for publication in principle.